# Physical Mapping of QTL in Four Spring Wheat Populations under Conventional and Organic Management Systems. I. Earliness

**DOI:** 10.3390/plants10050853

**Published:** 2021-04-23

**Authors:** Kassa Semagn, Muhammad Iqbal, Hua Chen, Enid Perez-Lara, Darcy H. Bemister, Rongrong Xiang, Jun Zou, Muhammad Asif, Atif Kamran, Amidou N’Diaye, Harpinder Randhawa, Curtis Pozniak, Dean Spaner

**Affiliations:** 1Department of Agricultural, Food, and Nutritional Science, 4-10 Agriculture-Forestry Centre, University of Alberta, Edmonton, AB T6G 2P5, Canada; fentaye@ualberta.ca (K.S.); mi1@ualberta.ca (M.I.); hchen@swust.edu.cn (H.C.); enid1@ualberta.ca (E.P.-L.); bemister@ualberta.ca (D.H.B.); rxiang@ualberta.ca (R.X.); jzou6@ualberta.ca (J.Z.); masif1@ualberta.ca (M.A.); dratifkamran1@gmail.com (A.K.); 2Department of Agronomy, School of Life Science and Engineering, Southwest University of Science and Technology, 59 Qinglong Road, Mianyang 621010, China; 3Department of Agronomy, 2004 Throckmorton Plant Science Center, Kansas State University, Manhattan, KS 66506, USA; 4Heartland Plant Innovations, Kansas Wheat Innovation Center, 1990 Kimball Avenue, Manhattan, KS 66502, USA; 5Seed Centre, Department of Botany, The University of Punjab, New Campus, Lahore 54590, Pakistan; 6Crop Development Centre, Department of Plant Sciences, University of Saskatchewan, 51 Campus Drive, Saskatoon, SK S7N 5A8, Canada; amidou.ndiaye@usask.ca (A.N.); curtis.pozniak@usask.ca (C.P.); 7Agriculture and Agri-Food Canada, 5403-1st Avenue South, Lethbridge, AB T1J 4B1, Canada; harpinder.randhawa@agr.gc.ca

**Keywords:** DArTseq, nitrogen use efficiency, mapping, organic agriculture, physical map, SNP array, wheat

## Abstract

In previous studies, we reported quantitative trait loci (QTL) associated with the heading, flowering, and maturity time in four hard red spring wheat recombinant inbred line (RIL) populations but the results are scattered in population-specific genetic maps, which is challenging to exploit efficiently in breeding. Here, we mapped and characterized QTL associated with these three earliness traits using the International Wheat Genome Sequencing Consortium (IWGSC) RefSeq v2.0 physical map. Our data consisted of (i) 6526 single nucleotide polymorphisms (SNPs) and two traits evaluated at five conventionally managed environments in the ‘Cutler’ × ‘AC Barrie’ population; (ii) 3158 SNPs and two traits evaluated across three organic and seven conventional managements in the ‘Attila’ × ‘CDC Go’ population; (iii) 5731 SilicoDArT and SNP markers and the three traits evaluated at four conventional and organic management systems in the ‘Peace’ × ‘Carberry’ population; and (iv) 1058 SNPs and two traits evaluated across two conventionally and organically managed environments in the ‘Peace’ × ‘CDC Stanley’ population. Using composite interval mapping, the phenotypic data across all environments, and the IWGSC RefSeq v2.0 physical maps, we identified a total of 44 QTL associated with days to heading (11), flowering (10), and maturity (23). Fifteen of the 44 QTL were common to both conventional and organic management systems, and the remaining QTL were specific to either the conventional (21) or organic (8) management systems. Some QTL harbor known genes, including the *Vrn-A1*, *Vrn-B1*, *Rht-A1*, and *Rht-B1* that regulate photoperiodism, flowering time, and plant height in wheat, which lays a solid basis for cloning and further characterization.

## 1. Introduction

The application of nitrogen (N), phosphorus (P), and potassium (K) nutrients through the application of chemical fertilizers have become a standard conventional practice for increasing yield and yield components. However, excess use of chemical fertilizers under the conventional management system (high-N) not only increases the costs of crop production but also reduces nitrogen use efficiency (NUE) and causes environmental damage, which is shifting the wheat (*Triticum aestivum* L.) breeding programs towards low input agricultural practices [1]. Organic production (low-N) is a low input agricultural practice that relies on crop rotation, mixed cropping, biological pest control, and fertilizers of organic origin, such as compost, manure, and green manure [2]. Although organic production has been an effective approach to achieve agricultural sustainability in the long term and its demand has increased in the last decade for different reasons [3,4,5], most current cultivars have shown poor performance and 5–31% lower grain yield as compared to the conventional management system [6]. Hence, there is a need to identify and/or develop genotypes with higher input use efficiency that produce high yield under optimal-N without yield penalty under low-N, and also discover genes and QTL associated with high-N and low-N management systems.

The wheat breeding group at the University of Alberta has been conducting extensive research in the Canada Western Red Spring (CWRS) wheat class, including (i) developing several improved cultivars [7,8,9,10], (ii) evaluating the phenotypic performance of diverse cultivars under conventional and/or organic managements [11,12,13,14,15,16], (iii) understanding the genetics of earliness that serves as baseline data for developing early maturing cultivars to avoid frost damage [17,18,19,20], and (iv) mapping genes and QTL associated with diverse traits using biparental populations [19,21,22,23,24,25,26,27] and a genomewide association mapping panel [28,29]. The development of early-maturing CWRS cultivars is of paramount importance in the northern breeding programs to provide farmers not only an option of growing the crop with minimal loss due to frost but also help to escape from the late incidence of diseases, heat, and drought as compared with their late-maturing counterparts. One of our aims was understanding the genetics of earliness under organic and conventional management systems, which is the basis for developing early maturing cultivars. In one of our studies, we evaluated the ‘Cutler’ × ‘AC Barrie’ recombinant inbred line (RIL) population across five conventionally managed field conditions and two greenhouse experiments and genotyped with 488 simple sequence repeat (SSR) and diversity arrays (DArT) markers. Using the phenotype data across all environments and a genetic linkage map, we uncovered two QTL on chromosome 1B and 5B associated with both flowering and maturity [19]. In a follow-up study, we reanalyzed the phenotype data of the ‘Cutler’ × ‘AC Barrie’ RIL population with a linkage map of 1809 SNPs selected out of the wheat 90K iSelect array and two functional markers (*Ppd-D1* and *Rht-D1*). That study uncovered 19 QTL associated with four traits, including six QTL for flowering and five QTL for maturity [23].

In another study, we evaluated the ‘Attila’ × ‘CDC Go’ RIL population at three conventionally and organically managed environments and genotyped the population with Diversity Arrays Technology (DArT) and the *Rht-B1* gene-specific functional markers. Using the phenotypic data across three environments and a linkage map of 580 markers, we uncovered three and five QTL associated with five agronomic traits under conventional and organic management systems, respectively [27]. For flowering and maturity, we only found environment (trial) specific QTL. To investigate if an increase in marker density improves QTL detection, we reanalyzed the same phenotype data of the eight traits averaged over the three organically managed environments and linkage maps of 1200 informative SNPs selected out of the wheat 90K iSelect array and three functional markers (*Ppd-D1*, *Vrn-A1*, and *Rht-B1*). Our analyses identified 16 QTL distributed across ten chromosomes [24] of which three were associated with flowering (1) and maturity (2). In conventional management, we evaluated the ‘Attila’ × ‘CDC GO’ RIL population for seven years and conducted QTL analysis using a linkage map of the 1203 markers. That study uncovered 14 QTL associated with eight traits, of which the three QTL for flowering and maturity remained the same as those identified in the organic management system [25].

Recently, we evaluated a RIL population derived from ‘Peace’ × ‘Carberry’ under two conventional and organic managements (2016–2017) and genotyped with DArT-based genotyping by sequencing (DArTseq) technology. Using the phenotype data of two environments per management, we found 53 QTL associated with nine agronomic traits, of which nine QTL were for heading (3) and maturity (6). Fourteen of the 53 QTL were common in both conventional and organic management systems [21]. We also mapped QTL associated with ten agronomic and end-use quality traits in ‘Peace’ × ‘CDC Stanley’ RIL population evaluated for two years (2016–2017) under conventional and organic management systems. Using phenotypic data across two environments per management and the IWGSC RefSeq v2.0 physical map position of 1058 informative SNPs selected out of the wheat 90K iSelect array, we uncovered a total of 27 QTL associated with nine traits, including QTL for heading (2) and maturity (4) [30]. Although we reported several QTL associated with the heading, flowering, and maturity time in the four mapping populations and two management systems, direct comparisons of the QTL results across independent studies was challenging due to the use of population-specific genetic maps, which form the basis of the present study.

Meta-QTL analysis [31] has been widely used for combining data from independent QTL mapping studies, identify the occurrence of QTL hotspots in a consensus genetic linkage map, and narrowing the QTL genetic confidence intervals [32,33,34,35,36,37]. QTL meta-analysis has been performed in wheat for several traits, including heading and maturity [36,38,39]. Some of the challenges in the use of both population-specific and consensus genetic maps for QTL detection in polyploid species, such as wheat, include (1) difficulty in establishing linkage groups and determining correct markers orders, (2) an increase in the number of linkage groups as compared with the number of chromosomes, which could be 2–3-fold [38], (3) collinearity among multiple markers due to relatively small population sizes that result to very small inter-marker interval (<1 cM) between multiple adjacent markers, and (4) the lack of consistent relationship between genetic and physical maps, which makes direct comparisons of QTL discovery results across multiple populations unreliable [39]. Although some studies have published physical maps of QTL associated with diverse traits [39,40,41,42,43,44], the method has not yet been widely used due to a lack of reliable physical maps. Recently, the International Wheat Genome Sequencing Consortium (IWGSC) has released the latest version of the bread wheat reference genome sequence (RefSeq v2.0) and physical map (http://wheat-urgi.versailles.inra.fr/; accessed on 22 April 2021). A simple online tool that helps breeders to map the chromosomal location and position of the wheat 90K iSelect array and genotyping by sequencing markers onto the flow-sorted Chinese Spring (CS) survey sequences have also been made available to the public (https://download.txgen.tamu.edu/shichen/mapper_v2.html; accessed on 22 April 2021). The objectives of the present study were, therefore, to (1) map and characterize QTL associated with heading, flowering and maturity time in four RIL populations using the IWGSC RefSeq v2.0 physical maps, and (2) assess the consistency of QTL identified in the conventional and organic management systems.

## 2. Results

### 2.1. Phenotype and Genotype Data

Our study was based on a total of 698 RILs from four biparental populations, which were evaluated under conventional and organic management systems and genotyped either with the wheat 90K iSelect SNP array or the DArTseq technology (Table 1). In both conventional and organic management systems, the seven parents used in developing the four RIL mapping populations on average differed by 4–5 days in heading, 7–8 days in flowering, and 8–9 days in maturity. In three of our mapping populations, (i) ‘CDC Go’ headed/flowered and matured 3–4 days earlier than ‘Attila’ both under conventional and organic management systems: (ii) ‘Cutler’ headed/flowered and matured three days earlier than ‘AC Barrie’ under conventional management; and (iii) ‘Peace’ headed/flowed a day and matured three days earlier than ‘CDC Stanley’ under both management systems. On the other hand, ‘Carberry’ headed/flowered 3–4 days earlier but matured 2–3 days later than ‘Peace’ in both management systems.

The detailed phenotypic performance for the ‘Attila’ × ‘CDC Go’ (hereinafter abbreviated as ACG) [24,25], the ‘Cutler’ × ‘AC Barrie’ (hereinafter abbreviated as CAB) [23], and ‘Peace’ × ‘CDC Stanley’ (hereinafter abbreviated as PCS) [30] RIL populations have been described in our previous studies. Figure 1 shows the frequency distribution of the three traits, and Table 2 provides a summary of the descriptive statistics and broad-sense heritability under conventional and/or organic management systems. Among the four RIL populations, the means of days to heading, flowering, and maturity ranged from 49 to 52 days, from 52 to 57 days, and from 88 to 98 days, respectively. Broad-sense heritability under conventional and organic management systems varied from 0.40 to 0.77 (Table 2). There was a highly significant (*p* < 0. 01) positive correlation between the organic and conventional management systems with regression coefficients varying from 0.71 to 0.86 (Figure 2).

Of the wheat 90K iSelect array used for genotyping the RIL populations in the ACG, CAB, and PCS populations, we constructed three independent genetic maps that consisted of 3158, 6526, and 1058 markers, respectively. The genetic map of the PAC population consisted of 5731 DArTseq markers, which included 3840 SilicoDArT and 1891 SNPs (Table 3 and Appendix A). The genetic map lengths in the ACG, CAB, PAC, and PCS populations were 1734, 3596, 25508, and 4922 cM, respectively. The corresponding IWGSC RefSeq v2.0 physical map length in the ACG, CAB, PAC, and PCS populations were 11859, 13788, 13740, and 13881 Mb, respectively. Overall, a total of 15183 markers were used in the genetic and physical maps of the four populations, of which 1204 and 43 markers were common in two and three of the four populations, respectively (Appendix A). Pearson correlation coefficients between the genetic and physical maps varied from 0.39 in the PAC to 0.60 in the CAB population (Appendix A), suggesting a low to moderate agreement between genetic and physical maps that ultimately affects QTL detection.

### 2.2. Physical Map of QTL

Using the least-squares means phenotype data and the IWGSC RefSeq v2.0 physical maps, we uncovered 36 QTL under the conventional management and 23 QTL under the organic management system (Appendix A). When the results from the two management systems were combined, we found a total of 44 QTL in the ACG (3), PCS (8), CAB (10), and PC (23). Fifteen of the 44 QTL were common to both conventional and organic management systems, with the remaining QTL being specific to either the conventional (21) or organic (8) management systems (Table 4 and Appendix A). All QTL were population specific (i.e., none of the QTL was detected in more than one population). For heading, we found a total of 11 QTL across seven chromosomes, which includes 8 QTL in the PAC and 3 QTL in the PCS populations. The 11 QTL for heading were mapped at 325.9–336.3 Mb on chromosome 1D (QHd.dms-1D), 55.8–62.7 Mb on 2B (QHd.dms-2B), at 45.2–49.0 Mb on 3B (QHd.dms-3B), at 17.7–17.8 Mb on 4A (QHd.dms-4A), both at 60.3–76.7 Mb (QHd.dms-5A.1) and at 620.5–621.3 Mb (QHd.dms-5A.2) on 5A, at 349.8–354.8 Mb (QHd.dms-5B.1), 396.8–400.7 Mb (QHd.dms-5B.2), 574.5–577.0 Mb (QHd.dms-5B.3) and 639.0–653.9 Mb (QHd.dms-5B.4) on 5B, and at 73.3–84 Mb on 7D (QHd.dms-7D). The QTL physical positions provided here, and subsequent sections are confidence intervals based on the left and right flanking markers instead of the exact position generated by the software. Each QTL individually explained from 1.8 to 19.3% and together accounted for 7.7–16.9% and 52.3–59.4% of the phenotypic variance for heading in the PCS and PAC populations, respectively (Appendix A). Five of the heading QTL (QHd.dms-2B, QHd.dms-5A.1, QHd.dms-5B.3, QHd.dms-5B.4, and QHd.dms-7D) were common in both organic and conventional management systems (Figure 3).

For flowering (anthesis), we found a total of ten QTL across seven chromosomes in the ACG (1 QTL), CAB (3), and PAC (6) populations (Figure 3, Table 4). The ten flowering QTL were mapped at 37.5–37.9 Mb (QFlt.dms-2A.1) and 700.6–700.9 Mb (QFlt.dms-2A.2) on chromosome 2A, at 55.8–62.7 Mb on 2B (QFlt.dms-2B), at 432.4–432.8 Mb (QFlt.dms-3B), at 582.8–583.0 Mb (QFlt.dms-5A.1) and 587.3–587.4 (QFlt.dms-5A.2) on 5A, at 333.3–349.8 Mb (QFlt.dms-5B.1) and 574.5–577.0 Mb (QFlt.dms-5B.2) on 5B, at 24.2–25.4 Mb on 7A (QFlt.dms-7A), and at 73.3–84.0 Mb on 7D (QFlt.dms-7D). Four of the flowering QTL (QFlt.dms-2B, QFlt.dms-5A.2, QFlt.dms-5B.2, and QFlt.dms-7D) were identified in both organic and conventional management systems. Each QTL individually explained from 2.0% to 20.8% and together accounted for 19.2%–37.8% of the phenotypic variance per population per management (Appendix A).

The twenty-three QTL for maturity were located across ten chromosomes (Figure 3, Table 4), which were identified in the ACG (2), PCS (5), CAB (7), and PAC (9) populations. They were mapped at 35.6–39.8 Mb (QMat.dms-1A.1), 399.4–426.6 Mb (QMat.dms-1A.2) and 443.1–462.8 Mb (QMat.dms-1A.3) on chromosome 1A; at 392.0–406.6 Mb (QMat.dms-3A.1) and 513.9–553.1 Mb (QMat.dms-3A.2) on 3A; at 45.2–49.0 Mb (QMat.dms-3B.1) and 821.1–835.0 Mb (QMat.dms-3B.2) on 3B; at 16.1–17.7 Mb (QMat.dms-4A.1), 65.5–68.5 Mb (QMat.dms-4A.2) and 580.4–582.7 Mb (QMat.dms-4A.3) on 4A; at 30.5–32.0 Mb (QMat.dms-4B.1) and 569.2–599.6 Mb (QMat.dms-4B.2) on 4B; at 21.0–33.0 Mb on 4D (QMat.dms-4D); at 569.9–570.1 Mb (QMat.dms-5A.1), 587.3–587.4 Mb (QMat.dms-5A.2), 619.7–620.2 Mb (QMat.dms-5A.3), and 689.1–692.6 Mb (QMat.dms-5A.4) on 5A; at 559.9–560.8 Mb (QMat.dms-5B.1) and 574.5–577.0 Mb (QMat.dms-5B.2) on 5B; at 96.8–110.0 Mb (QMat.dms-7A.1), 133.5–134.2 Mb (QMat.dms-7A.2), and 680.7–717.9 Mb (QMat.dms-7A.3) on 7A, and at 73.3–84.0 Mb on 7D (QMat.dms-7D) (Table 4 and Appendix A). Six of the maturity QTL (QMat.dms-4B.1, QMat.dms-5A.2, Qmat.dms-5A.3, QMat.dms-5A.4, QMat.dms-7A.2, and QMat.dms-7D) were detected in both organic and conventional management systems. Each maturity QTL individually explained from 1.7% to 16.7% and together accounted for 4.0%–33.4% of the phenotypic variance per population per management (Appendix A).

### 2.3. Coincident QTL

Overall, we found eight genomic regions on chromosomes 2B, 3B, 4A, 5A, 5B, and 7D harboring QTL clusters for 2–3 traits within each population (Figure 3, Table 4, and Appendix A). One of the coincident QTL spans 6.9 Mb between 55.8 and 62.7 Mb on chromosome 2B, which is associated with heading (QHd.dms-2B) and flowering (QFlt.dms-2B) in the PAC population in both conventional and organic management systems. The second and third coincident QTL were associated with days to heading and maturity in the PCS population both on chromosomes 3B (45.2–49.0 Mb) and 4A (16.1–17.8 Mb). The fourth and fifth coincident QTL were associated with days to heading, flowering, and maturity in the PAC population evaluated in both management systems; these QTL, and they were mapped on chromosomes 5B (574.5–577.0 Mb), and 7D (73.3–84.0 Mb). The remaining three coincidental QTL were located on chromosome 5A that were associated with days to heading and maturity in the PAC population (619.7–621.3 Mb), flowering and maturity in the ACG population (587.3–587.4 Mb), and on chromosome 5B for heading and flowering in the PAC population (333.8–354.8 Mb). The long arms of chromosomes 5A and 5B consisted of QTL clusters from three (PAC, ACG, and CAB) and two (PAC and PCS) populations, respectively (Figure 3).

## 3. Discussion

### 3.1. QTL Based on Genetic and Physical Maps

In our previous study [23], QTL mapping conducted in the ‘Cutler’ and ‘AC Barrie’ population using genetic maps of 1811 markers and flowering and maturity data under field conditions converted into growing degree days (GDD) identified six QTL for flowering (2D, 3B, 4A, 5A, 6B and 7A) and five QTL for maturity (2D, 4A, 4D and 7A), which individually explained between 6.5 and 25.4% of the phenotypic variance under a conventional management system. However, we found out that the chromosomal location of flanking SNP markers of four of the QTL reported in our previous study differed from the physical map. When analyses were done using days to flowering and maturity time without conversion to GDD, we found four QTL for flowering on 2D, 3B, 6B, and 7A, and four QTL for maturity on 2D, 4A, 4D, and 7A (Appendix A). In the present study conducted using the IWGSC RefSeq v2.0 physical map and days to flowering and maturity data, we identified 3 QTL for flowering on chromosomes 3B, 5A, and 7A, and seven QTL for maturity on 1A, 3B, 4A, 4D, and 5A (Figure 3, Appendix A). Although chromosomes 7A for days to flowering and 4A, 4D, and 5A for days to maturity were common in both our previous and the present studies, their positions are different. The left and right flanking markers identified in our previous study were physically far apart, including some on different arms of the same chromosome. QTL mapping conducted in the ‘Attila’ × ‘CDC Go’ RIL population using genetic maps of 1203 markers identified a coincident genomic region associated with both flowering and maturity under both management systems on chromosome 5A and another region on 4B associated with maturity under both management systems [24,25]. Each of these two genomic regions accounted for 5.9%–17.2% of the phenotypic variance per trait per management system (Appendix A). QTL mapping conducted using the IWGSC RefSeq v2.0 physical map in this study identified both genomic regions, each individually explaining 3.7%–20.8% of the phenotypic variances for flowering and maturity across all environments (Appendix A).

Using the phenotype data of ‘Peace’ and ‘Carberry’ RIL populations evaluated for two years under conventional and organic management systems and linkage map of 4439 markers generated via DArTseq technology, we previously [21] uncovered three QTL on chromosomes 1A, 5B, and 7D for heading and six QTL on 3A, 4A, 4B, 4D, 5B and 7D for maturity (Appendix A). In the present study conducted using the IWGSC RefSeq v2.0 and four years phenotype, we identified eight QTL for heading and nine QTL for maturity (Figure 3, Table 4). Seven of the nine QTL (all except 1A for heading and 4D for maturity) reported in our previous study were identified in the present study. Moreover, we uncovered new QTL on chromosomes 1D and 2B for heading and chromosome 5A for both heading and maturity. It should, however, be noted that the phenotype data used in the present study had four environments (2016–2020) instead of the two years (2016–2017) used in our previous study. In the ‘Peace’ and ‘CDC Stanley” RIL population, QTL analyses in the previous [30] and present study were done using the IWGSC RefSeq v2.0 physical maps, so all QTL identified in our previous study were also identified in the present study. Because of additional data cleaning, however, two new QTL were uncovered in the present study on 4A (17.6–17.7 Mb) for heading and on 7A (96.8–110.0 Mb) for maturity. Because of the overlap in the IWGSC RefSeq v2.0 physical confidence interval among some of the QTL and/or presence of eight coincident genomic regions associated with two to three traits, the 44 QTL (Figure 3) identified in the present study fell into 31 genomic regions (Appendix A). Nearly a third of the QTL were located on homoeologous Group 5 chromosomes (15 QTL), followed by Groups 3, 4, and 7 (seven QTL each), Group 2 (five QTL), and Group 1 (four QTL). Hanocq et al. [36] summarized 177 QTL reported across 13 independent studies on the D. Somers wheat consensus map [45]. After projecting 84 of the 177 initial QTL on the consensus map, the authors reported 30 genomic regions involved in the control of earliness (heading and flowering) and its three components (photoperiod sensitivity, vernalization requirement, and intrinsic earliness). Their meta-analysis demonstrated that the genetic control of earliness and its components involves not only the major *Ppd* and *Vrn* genes on homoeologous Groups 2 and 5 chromosomes, respectively, but also other genomic regions in Groups 4 and 7, which is evident in the current study.

Heading and flowering are important adaptative traits that determine the adaptation of wheat to diverse eco-geographical regions and are critical for yield potential and stability [46,47,48]. We uncovered genomic regions associated with heading on seven chromosomes (1D, 2B, 3B, 4A, 5A, 5B, and 7D) and for flowering on seven chromosomes (2A, 2B, 3B, 5A, 5B, 7A, and 7D) of which four genomic regions (55.8–62.7 Mb on 2B, 333.3–354.8 and 574.5–577.0 Mb on 5B, 73.3–84.0 Mb on 7D) were common for both traits (Table 4 and Appendix A). Using three RIL mapping populations evaluated across four environments, Zhao et al. [48] reported a total of 25 QTL associated with heading (15) and flowering (10) across 17 chromosomes, which individually explained from 4.3 to 32.5% of the phenotypic variance. The authors reported QTL for heading on fifteen chromosomes, including 1D, 2B, 3B, 4A, 5A, and 5B. They also reported flowering QTL on ten chromosomes, including 2B and 5B.

Four of the thirty-one genomic regions identified in the present study were mapped on chromosome 5A (Figure 3, Appendix A), which were associated with heading (60.3–76.7 Mb), maturity (689.1–692.6 Mb), both flowering and maturity (569.8–587.5 Mb), and both heading and maturity (619.7–621.3 Mb). Each region consisted of 17–208 protein-coding candidate genes, including the *Vrn-A1* (TraesCS5A02G391700) gene at 587.4 Mb (5A: 587411454–587423416) and a flowering-promoting factor-1 gene (TraesCS5A02G530200) at 5A: 689155443–689156097 (https://plants.ensembl.org/index.html; accessed on 22 April 2021). Chromosome 5B consisted of four genomic regions associated with both heading and flowering (333.8–354.8 Mb), heading (396.8–400.7 and 639.0–653.9 Mb), and all three traits (559.8–577.0 Mb). The 639.0–653.9 Mb interval consisted of 165 candidate genes, including TraesCS5B02G481200 at 653.8 Mb (5B:653794081–653798305 bp), which plays an important role in regulating flowering, photoperiodism, and mRNA splicing. The 559.8–577.0 Mb interval associated with a heading, flowering, and maturity consists of a total of 174 protein-coding candidate genes, including the *Vrn-B1* (TraesCS5B02G396600) at 573.8 Mb (5B:573802883–573816070).

The genetic control of earliness in wheat is controlled primarily by the photoperiod response (*Ppd*), vernalization requirement (*Vrn*), and ‘earliness per se’ (*Eps*) genes. *Eps* genes act independently of the *Vrn* and *Ppd* genes [47,49]. Photoperiod insensitivity is controlled primarily by the dominant alleles at *Ppd-A1*, *Ppd-B1*, and *Ppd-D1* that are located on chromosomes 2A, 2B, and 2D, respectively [50,51,52,53]. Spring wheat introgression lines lacking *Ppd-B1* flowered 10–15 days later than controls under long days as compared to 1–5 days for lines that did not have the *Ppd-A1* allele; loss of *Ppd-D1* alleles did not affect flowering time [54]. On homologous Group 2 chromosomes, we found two regions on 2A (32.4–37.9 and 700.6–700.9 Mb) for flowering and one region on 2B (55.8–62.7 Mb) for both heading and flowering (Figure 3). The two genomic regions for flowering on chromosome 2A consisted of three candidate genes, while the single genomic region on chromosome 2B consisted of 60 candidate genes. Although we are not sure about the exact physical positions of *Ppd-A1* and *Ppd-B1* due to conflicting positions of the SSR markers linked to both genes, the genomic regions that we identified on both 2A and 2B may contain the two photoperiod response genes.

The three genomic regions uncovered on 4A were associated with heading and maturity (16.0–17.7 Mb) and maturity (both at 65.4–68.5 Mb and 580.0–582.7 Mb). Each genomic region on chromosome 4A consisted of 31–35 protein-coding candidate genes, including the *Rht-A1* gene (TraesCS4A02G271000) at 582.5 Mb (4A:582477351–582479578). The two genomic regions that we identified on 4B (30.5–32.0 and 569.2–599.6 Mb) were associated with maturity, which consisted of 11 and 225 protein-coding candidate genes, respectively, including the *Rht-B1* (TraesCS4B02G043100) at 4B:30861268–30863723, TraesCS4B02G308800 (4B:599399570–599399905) and TraesCS4B02G308700 (4B:599356666–599356977); both TraesCS4B02G308800 and TraesCS4B02G308700 are flower promoting factors. Some of the dwarfing genes, such as *Rht-B1*, *Rht5*, *Rht8*, and *Rht12* have been reported in slightly delaying heading, flowering, and/or maturity time in wheat [55,56,57]. The single genomic region on 4D (21.0–33.0 Mb) was associated with maturity, which harbors clusters of 123 candidate genes of unknown function.

On chromosome 7A, we found one genomic region associated with flowering (24.2–25.4 Mb) and three genomic regions associated with maturity (at 96.8–110.0, 133.4–134.2, and 680.6–717.9 Mb). These regions harbor between 10 and 473 genes, including the TraesCS7A02G146100 (7A:97873101–97873865) that plays a role in the signaling pathway. On chromosome 7D, we identified a single genomic region between 73.3 and 84.0 Mb associated with a heading, flowering, and maturity. This region consisted of 138 candidate genes, including TraesCS7D02G123900 (7D:77241990–77242100), which is Photosystem II reaction center protein I. Both chromosomes 1A and 3A consisted of three (35.5–39.8, 399.4–426.6, and 443.0–462.8 Mb) and two (392.0–427.8 and 513.8–553.1 Mb) genomic regions, respectively, which all were associated with maturity. Each genomic region on chromosome 1A and 3A harbors clusters of protein-coding candidate genes ranging from 47 to 205 (Appendix A), but their function has not yet been established. The interval from 325.8 to 336.3 Mb on chromosome 1D was associated with heading, which consists of 87 candidate genes of unknown function. On chromosome 3B, we found three genomic regions associated with heading and maturity (45.2–49.0 Mb), flowering (432.4–432.8 Mb), and maturity (821.1–835.0 Mb). The intervals 45.2–49.0 and 821.1–835.0 Mb contained 36 and 123 candidate genes, respectively, whereas the other interval associated with flowering harbors no candidate gene.

### 3.2. Effects of Genetic Background and Management

All QTL identified in the present study were population (cross) specific. The closest QTL detected in this study were QFlt.dms-5A.1 in the CAB population and both QFlt.dms-5A.2 and QMat.dms-5A.2 in the ACG population (Figure 3), but there is a ~4.6 Mb interval between them. Although identification of QTL conserved across multiple genetic backgrounds is one of the prerequisites for marker-assisted selection, most QTL reported in the literature are population (genetic background) specific [37,58,59,60,61,62], which restricts their application for predicting phenotypic performance across diverse genetic backgrounds. Brasier et al. [62] evaluated two biparental winter wheat mapping populations derived from a cross between two high NUE parents and a shared common low NUE parent for 11 traits. Using genetic maps of 3147 and 3918 markers, the authors uncovered a total of 130 QTL, of which 10 QTL were common between the two populations. Symonds et al. [60] detected nine QTL associated with trichome density in four RIL populations of *Arabidopsis thaliana* (two of their populations shared a common parent), of which only two QTL were detected in all four populations; the other 7 QTL were population specific. In rice, Yao et al. [58] evaluated three biparental populations with a common parent and uncovered 28 QTL associated with African rice gall midge resistance, but there was only a single QTL common in two of the three populations. In soybean, Kang et al. [61] used two mapping populations with a common parent and uncovered eight QTL associated with pod dehiscence, but only one QTL was common between the two populations. Semagn et al. [37] used 18 bi-parental maize populations and identified a total of 183 QTL associated with grain yield (101) and anthesis silking interval (82) under drought and irrigated conditions. However, only a few QTL were detected across 2–6 populations and/or the two water regimes.

Identification of consistent QTL in both conventional (high-N) and organic (low-N) management systems would be highly useful for improving spring wheat through marker-assisted selection. However, our results demonstrated that ~34% of the 44 QTL uncovered in the present study were common between the organic and conventional management systems, with the remaining being specific to the conventional (48%) or organic (18%) management systems (Figure 3, Table 4). QTL detected in both management systems would be ideal for developing improved wheat germplasm using marker-assisted breeding irrespective of management system, while those QTL detected in the conventional (high-N) or organic (low-N) managements should be considered in their respective managements. Several studies have reported similar results in diverse crops, including wheat [62,63,64,65,66], rice [67,68], barley [69], sorghum [70], and potato [71]. Using a doubled haploid winter wheat mapping population evaluated at two N fertilizer treatments under field conditions, An et al. [64] detected a total of 17 QTL associated with different traits under low-N (9) and high-N (8), but none were detected in both N treatments. Laperche et al. [66] evaluated grain protein, grain yield, and their components in a DH winter wheat population under high-N and low-N, which detected a total of 67 QTL under high-N and 51 QTL under low-N, of which 13 QTL were detected at both N levels. Yue et al. [68] evaluated a rice RIL mapping population under normal-N and low-N levels and detected a total of 52 QTL associated with four yield-related traits. Eleven of the 52 QTL were detected under the two N levels and the remaining 30 QTL were detected either in the normal-N (17) or low-N (13).

Using a RIL sorghum mapping population evaluated under normal-N and low-N fertilizer conditions, Gelli et al. [70] detected a total of 38 QTL associated with 11 agronomic traits, of which four QTL were common between the two N management systems, and the remaining were specific either to the low-N (16) or normal-N (14). RNA sequencing analyses on sorghum seedling root tissues revealed 726 differentially expressed gene transcripts related to nitrogen uptake and metabolism between parents, of which 108 were mapped close to the QTL regions. Expression analysis of N metabolism-related genes reveals differences in the performance of two barley cultivars that were adapted to low-N and high-N levels [69]. The authors reported that some N metabolism-related genes were only induced in shoots of low-N tolerant varieties [69,71]. Leaf and root transcriptomic profiles analyses in potato cultivars have also revealed differentially expressed N metabolism-related genes [71], which may be one possible reason for the mapping of multiple management-specific QTL in our study.

## 4. Materials and Methods

### 4.1. Phenotyping and Genotyping

The present study was conducted on four RIL populations derived from crosses involving ‘Cutler’ [72], ‘AC Barrie’ [73], ‘Peace’ [74], ‘Carberry’ [75], ‘Attila’ [76], ‘CDC Stanley’ (https://www.inspection.gc.ca/english/plaveg/pbrpov/cropreport/whe/app00007708e.shtml; accessed on 22 April 2021), and ‘CDC Go’ (https://www.grainscanada.gc.ca/en/grain-quality/variety-lists/2020/2020-19.html; accessed on 22 April 2021). ‘Attila’ was bred by the International Maize and Wheat Improvement Center [76] and the other six parents were spring wheat cultivars bred in Canada. These parents have different combinations of alleles at the *Vrn1* and *Ppd-D1* genes [77,78]. ‘AC Barrie’, ‘CDC Go’, and ‘Carberry’ have the photoperiod sensitive *Ppd-D1b* alleles, while ‘Cutler’ and ‘CDC Stanley’ has the photoperiod insensitive *Ppd-D1a* allele [77].

As summarized in Table 1, we evaluated a total of 698 RILs representing the CAB (158), PCS (165 RILs), ACG (167), and PAC (208) populations. The 158 RILs derived from the CAB population and the two parents were evaluated at five environments under conventionally managed field conditions [19,23]. ‘AC Barrie’ [73] is a late-maturing cultivar compared to ‘Cutler’ [72] and carries the dominant *Vrn-A1a*, *Vrn-B1* and *Vrn-D1* vernalization alleles [77]. ‘Cutler’ has the dominant *Vrn-A1a* and the recessive *vrn-B1* and *vrn-D1* vernalization alleles. The 167 RILs from the ACG population and their parents were phenotyped at three organically managed [24,27], and seven conventionally managed [25] field conditions. ‘Attila’ is an early maturing cultivar as compared to ‘CDC Go’. The 208 RILs from the PAC populations and their parents were evaluated at four conventionally and organically managed field conditions in 2016–2020 as described in our recent study [21]. ‘Peace’ [74] is adapted to the shorter-season wheat-growing regions of the Canadian prairies and carries the dominant *Vrn-A1a* allele and the recessive *vrn-B1* and *vrn-D1* alleles [78], while ‘Carberry’ [75] carries the *Vrn-A1a*, and *Rht-B1b* alleles [77]. The 165 RILs from the PCS population and their parents were evaluated at two conventionally and organically managed fields in 2016–2017 [30]. ‘CDC Stanley’ is a medium maturing cultivar (https://www.inspection.gc.ca/english/plaveg/pbrpov/cropreport/whe/app00007708e.shtml; accessed on 22 April 2021).

The conventional and organic evaluation sites are located at the University of Alberta South Campus (53°19′ N, 113°35′ W), Edmonton, Canada, and were about 500 m apart. Trials were laid out in randomized incomplete block design, with two replications. Trials in the conventional management were planted earlier, broadcast-fertilized with 70 kg ha^−1^ of 46–0–0 (N–P_2_O_5_–K_2_O) in early spring, and band-fertilized during seeding with 36 kg ha^−1^ of 11–52–0 (N–P_2_O_5_–K_2_O). Weeds were controlled in conventional trials using registered herbicides following local recommendations and label directions. The trials under organic management were planted later to facilitate early season land operations (tillage) and did not receive any of the chemical inputs. The four-year crop rotation in the organic land was wheat, rye (*Secale* L.) plow-down, field pea (*Pisum sativum*), and canola (*Brassica napus* L.), whereas conventional land followed a three-year rotation of wheat, field pea, and canola.

The methodologies for DNA extraction and genotyping have been described in our previous studies in the CAB [23], ACG [24], PAC [26], and PCS [30] populations. DNA samples from the ACG, CAB, and PCS populations were genotyped at the University of Saskatchewan Wheat Genomics lab, Saskatoon, Canada, with the wheat 90K iSelect array that consisted of 81,587 SNPs [79], while the PAC population was genotyped with the DArTseq™ technology by DArT Pty Ltd., Canberra, Australia [80,81]. In addition, we also screened the two parents of each population if they were polymorphic for a few functional markers linked to photoperiod response (*Ppd-B1* and *Ppd-D1*) [82], vernalization response (*Vrn-A1* and *Vrn-B1*) [51], and height reducing *Rht-B1* [83] genes at the Agricultural Genomics and Proteomics Lab, University of Alberta, Edmonton, Canada, as described in our previous studies [21,23,53]. In cases where the two parents of each population showed polymorphism for the functional markers, we genotyped the RILs with the markers and used them for mapping.

### 4.2. Statistical Analysis

Least-squares means of each phenotypic trait and broad-sense heritability were computed using SAS version 9.4 (SAS Institute Inc., Cary, NC, USA) as described in our recent study [21]. Pearson correlations, regression (R^2^) coefficients, and different types of graphs were generated using both Minitab v14 (https://www.minitab.com; accessed on 22 April 2021) and JMP v7 (www.jmp.com; accessed on 22 April 2021) statistical software. For the PAC population, we received a total of 36,626 markers from DArT Pty Ltd., Canberra, Australia (https://www.diversityarrays.com; accessed on 22 April 2021), which consisted of 22,741 SilicoDArT markers (present vs. absent variation) and 13,885 SNPs along with 69 bp sequence based on the Chinese Spring (CS) reference sequence. Nearly 42% of the markers (11,704 SilicoDArT and 3557 SNPs) were polymorphic between the two parents. Marker sequences obtained from DArT Pty Ltd. were used for BLAST searches against the Chinese spring genome, the International Wheat Genome Sequence Consortium (IWGSC) RefSeq v1.0 [84] and RefSeq v2.0, which are available at http://download.txgen.tamu.edu/shichen/mapper_v2.html and http://wheat-urgi.versailles.inra.fr/; (accessed on 22 April 2021) The top hits with the highest alignment length, highest similarity (>95%), and/or expected value of 1 × 10^−20^ were used for physical mapping as described elsewhere [42]. Nearly 58% of the polymorphic markers (8,780 of the 15,261 markers) with physical information were selected for linkage analysis using MapDisto for Windows v2.1.7.10 [85]. After excluding all markers that showed segregation distortion at *p* < 0.01, a missing data of >20%, and those that were either unlinked or formed a linkage group with <5 markers, we retained 7066 markers for final map construction. Linkage groups were assigned to individual wheat chromosomes following the IWGSC RefSeq v2.0. Markers that were either assigned to a chromosome different from expected based on IWGSC RefSeq v2.0, had incorrect locus order, and/or that significantly affected the map length were further removed from the datasets. Final genetic linkage maps were constructed using the IWGSC RefSeq v2.0 marker order in MapDisto. Linkage analyses and genetic mapping in the other three populations (ACG, CAB, and PCS) genotyped with the wheat 90K iSelect array were done as described above. The only difference with the PAC population was the source of the sequence information for the wheat 90K iSelect array, which was retrieved from http://download.txgen.tamu.edu/shichen/flanking_v2.html (accessed on 22 April 2021).

Inclusive composite interval mapping (ICIM) was done on the least-squares means using QTL IciMapping version 4.2.53 [86] as described in our previous studies using the IWGSC RefSeq 2.0 physical maps with the following parameters: a minimum logarithm of the odds (LOD) scores of 3.0, a mean replacement for missing phenotypic data, an additive model to determine the effect of individual QTL, and a walking step of 100 kb. QTL scanning was performed independently on each population using population-specific physical maps, while QTL results were presented on to the integrated physical map of all four populations. QTL names were assigned by following the International Rules of Genetic Nomenclature (http://wheat.pw.usda.gov/ggpages/wgc/98/Intro.htm; accessed on 22 April 2021), which comprised of trait acronym, lab designation (dms = Dean Michael Spaner), and chromosome number. A physical QTL map of each chromosome was generated using MapChart v2.1 [87]. The start positions of the left and right flanking markers of each QTL (confidence interval) were used to search for candidate genes at the Ensemble Plants using *Triticum aestivum* genome (https://plants.ensembl.org/index.html; accessed on 22 April 2021). To minimize the length of the manuscript. However, the results from the Candidate genes search were used only in the discussion section.

## 5. Conclusions

Using the IWGSC RefSeq 2.0 physical map and phenotype data of four RIL mapping populations evaluated under conventional and organic management systems, we mapped and characterized a total of 44 QTL associated with a heading (11), flowering (10), and maturity (23) dates in hard red spring wheat. Fifteen of the 44 QTL were common in the two management systems, which would be ideal for developing improved wheat germplasm using marker-assisted breeding irrespective of management conditions. Those QTL detected under conventional (high-N) and organic (low-N) management would be useful for MAS in each management specific conditions. Each QTL individually explained from 1.7 to 20.8% and together accounted for 14.7%–59.4% and 4.0%–52.3% of the phenotypic variance under conventional and organic management systems, respectively. Because of the overlap in the physical confidence interval of some of the QTL and/or presence of eight coincident QTL associated with 2–3 traits, however, there are only 31 genomic regions associated with the three earliness traits. Some of the genomic regions harbor known genes, including the vernalization response *Vrn-A1* on chromosome 5A and *Vrn-B1* on 5B, as well as the height reducing *Rht-A1* on 4A and *Rht-B1* on 4B; these genes regulate photoperiodism, flowering time, and plant height in wheat. As far as we are aware, this is the first comprehensive physical map of genomic regions associated with the heading, flowering, and maturity in hard red spring wheat populations that would provide valuable information to wheat researchers and opens the opportunity for direct comparisons of QTL discovery studies across independent studies and possibly map-based cloning of a few regions.

## Figures and Tables

**Figure 1 plants-10-00853-f001:**
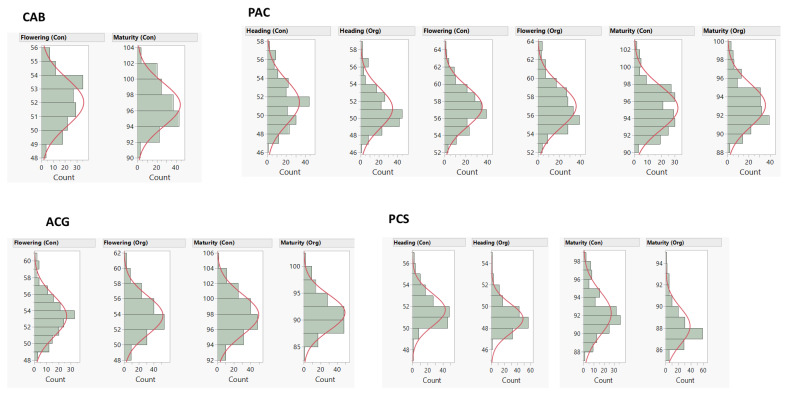
Frequency distribution of the least-squares means of days to heading, flowering, and maturity in four Canada Western Red Spring (CWRS) recombinant inbred line (RIL) mapping populations evaluated under conventional (Con) and organic (Org) management systems. The y-axes are days to heading (d), flowering (d), and maturity (d). Population code is as follows: ‘Cutler’ × ‘AC Barrie’ (CAB), ‘Attila’ × ‘CDC Go’ (ACG), ‘Peace’ × ‘Carberry ‘(PAC), and ‘Peace‘ × ‘CDC Stanley’ (PCS).

**Figure 2 plants-10-00853-f002:**
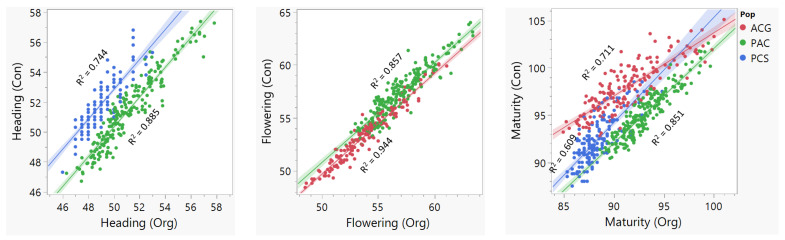
Scatter plots of the least squares means phenotype data under conventional (Con) and organic (Org) management systems. The ‘Cutler’ × ‘AC Barrie” population was evaluated only under a conventional management system and not included here. For each of the other three populations, the fitted lines and regression coefficients (R^2^) between the conventional and organic management systems are shown. Population codes—CAB: ‘Cutler’ × ‘AC Barrie’; ACG: ‘Attila’ × ‘CDC Go’; PAC: ‘Peace’ × ‘Carberry’; and PCS: ‘Peace’ × ‘CDC Stanley’.

**Figure 3 plants-10-00853-f003:**
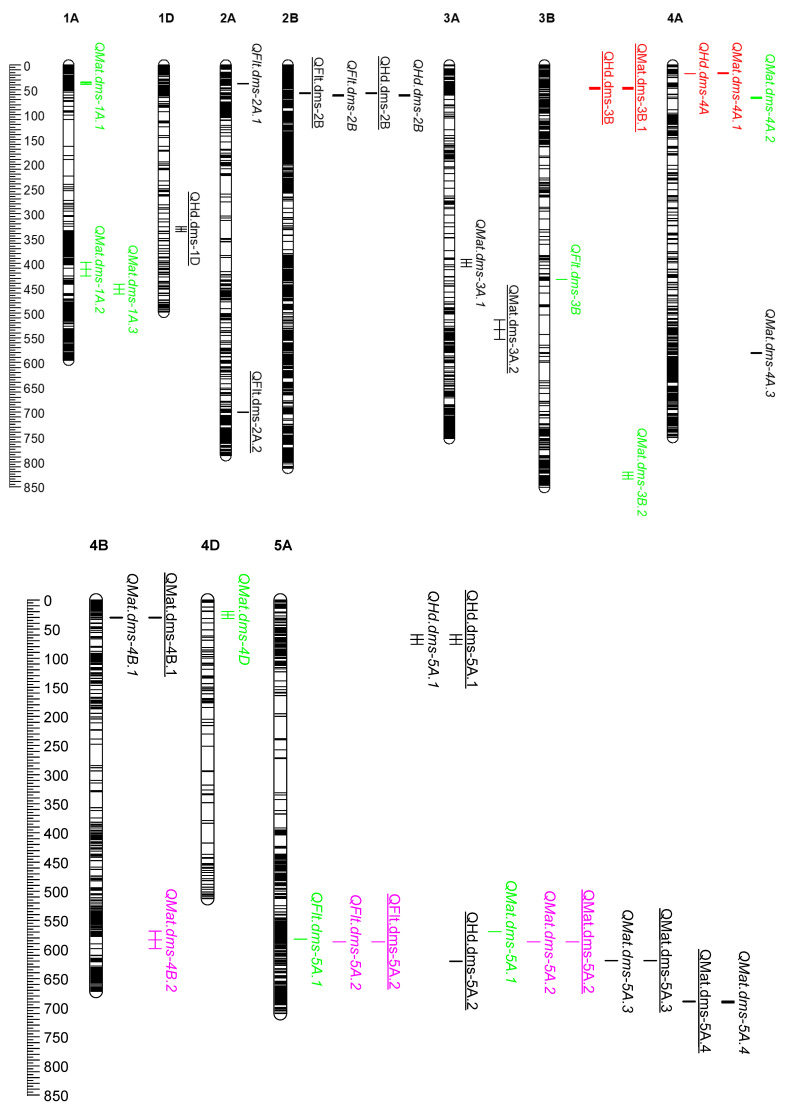
Physical map of quantitative trait loci (QTL) associated with days to heading, flowering, and maturity in four RIL populations evaluated under conventional and organic management systems: ‘Cutler’ × ‘AC Barrie’ (green), ‘Attila’ × ‘CDC Go’ (pink); ‘Peace’ × ‘Carberry’ (black), and ‘Peace’ × ‘CDC Stanley’ (red). The IWGSC RefSeq v2.0 physical map position (Mb) is shown on the left side of the chromosomes, with each horizontal line representing a marker. QTL are shown on the right side of each chromosome, with bars indicating their interval between the two flanking markers. QTL associated with conventional and organic managements are in italic and underlined, respectively. Details of the physical map and QTL information are given in Appendix A, respectively.

**Table 1 plants-10-00853-t001:** Summary of the four recombinant inbred line (RIL) populations used in the present study.

Population Code	Initial Cross	Population Size	Phenotyping	Genotyping *	References
CAB	‘Cutler’ × ‘AC Barrie’	158	Evaluated five times in 2007–2008 and 2010–2012 under the conventional management system.	Genotyped with the wheat 90K Illumina iSelect SNP array and two functional markers (*Ppd-D1* and *Rht-D1*).	[23]
ACG	‘Attila’ × ‘CDC Go’	167	Evaluated three times (2008–2010) under organic and seven times (2008–2014) under conventional management system.	Genotyped with the wheat 90K Illumina iSelect SNP array and three functional markers (*Ppd-D1*, *Vrn-A1* and *Rht-B1*).	[24,25]
PAC	‘Peace’ × ‘Carberry’	208	Evaluated four times from 2016 to 2020 under organic and conventional management systems.	Genotyped with 36,226 markers (22,741 SilicoDArT and 13,885 SNPs) using DArTseq-based genotyping by sequencing method and three functional markers (*Vrn-B3*, *Rht-B1*, and *Glu-A3*).	[21,26]
PCS	‘Peace’ × ‘CDC Stanley’	165	Evaluated twice (2016–2017) under organic and conventional management systems.	Genotyped with the wheat 90K Illumina iSelect SNP array and three SSR markers (DuPw004, barc170, and wmc650).	[30]

* The wheat 90K Illumina iSelect SNP array consisted of a total of 81,587 SNPs. See Appendix A for details of the SNPs used for QTL mapping.

**Table 2 plants-10-00853-t002:** Descriptive statistics and broad-sense heritability for heading, flowering, and maturity time in each of the four recombinant inbred line populations under conventional and organic management systems.

Trait	Management ^1^	‘Cutler’ × ‘AC Barrie’			‘Attila’ × ‘CDC Go’			‘Peace’ × ‘Carberry’			‘Peace’ × ‘CDC Stanley’		
		Mean	Range	H ^2^	Mean	Range	H ^2^	Mean	Range	H ^2^	Mean	Range	H ^2^
Heading (days)	Con							51.4	46.7–57.4	0.72	51.7	47.3–56.8	0.64
	Org							51.2	46.3–58.6	0.77	48.9	46.0–54.0	0.70
Flowering (days)	Con	52.0	48.3–55.7	0.43	53.3	48.4–60.1	0.73	57.3	52.4–64	0.75			
	Org				53.8	48.5–61.1	0.72	57.0	52.1–63.4	0.71			
Maturity (days)	Con	96.7	91.6–103.7	0.50	97.9	92.8–105.1	0.45	95.2	90.4–102.9	0.53	92.2	87.5–98.5	0.62
	Org				91.3	84.9–101	0.44	93.1	88.6–99.6	0.40	88.2	85.3–94.8	0.53

^1^ Con: Conventional; Org: Organic management. ^2^ Broad-sense heritability.

**Table 3 plants-10-00853-t003:** Summary of the number of markers, genetic and physical maps in each of the four mapping populations. The physical map is based on the International Wheat Genome Sequencing Consortium (IWGSC) RefSeq v2.0.

Chrom	‘Cutler’ × ‘AC Barrie’			‘Attila’ × ‘CDC Go’			‘Peace’ × ‘Carberry’			‘Peace’ × ‘CDC Stanley’		
	No. of Markers	Genetic Map Length (cM)	Physical Map Length (bp)	No. of Markers	Genetic Map Length (cM)	Physical Map Length (bp)	No. of Markers	Genetic Map Length (cM)	Physical Map Length (bp)	No. of Markers	Genetic Map Length (cM)	Physical Map Length (bp)
1A	302	129	596,457,062	273	78	595,208,262	371	535	595,860,808	59	147	593,443,946
1B	237	244	698,233,083	54	25	698,181,188	457	1485	692,480,911	111	538	683,188,703
1D	134	34	353,726,668	32	6	410,643,066	77	158	451,725,315	33	164	479,789,480
2A	353	321	787,699,648	255	115	786,363,784	269	1591	759,862,164	81	230	770,603,608
2B	869	567	795,242,489	391	163	665,646,566	347	181	450,448,794	119	371	812,709,904
2D	93	35	598,126,556	30	40	555,087,626	129	4234	654,057,050	29	143	645,341,017
3A	378	249	752,968,879	173	121	737,952,205	380	265	753,051,285	16	234	745,304,963
3B	444	268	851,724,030	80	224	851,724,030	241	5223	848,161,565	12	122	847,814,062
3D	25	28	579,990,391	-	-	-	119	176	617,655,900	9	31	606,042,121
4A	424	180	725,639,518	194	62	681,910,513	222	1845	751,532,475	45	158	722,926,434
4B	320	85	669,758,748	44	42	655,593,292	210	125	672,766,284	38	195	644,427,716
4D	35	31	482,822,161	-	-	-	45	63	513,418,898	15	88	456,267,708
5A	415	132	710,849,404	124	106	667,797,441	223	414	705,303,717	64	454	684,682,180
5B	957	626	714,558,337	269	292	697,707,787	579	523	713,404,985	147	172	714,258,884
5D	16	4	557,692,932	38	3	569,677,699	160	4049	568,959,847	15	407	568,666,242
6A	374	94	601,826,399	169	59	498,653,429	275	395	622,068,866	58	140	619,171,526
6B	187	66	730,597,670	525	151	719,416,487	423	1002	731,066,251	42	60	729,835,185
6D	32	16	494,665,522	38	7	7,562,690	130	177	492,458,353	15	317	494,670,623
7A	559	166	744,464,513	252	130	744,464,355	602	1999	743,593,777	86	351	733,173,741
7B	338	290	763,315,508	200	92	739,367,419	334	496	759,405,691	57	531	722,136,838
7D	34	33	577,952,590	17	18	575,785,725	138	573	642,704,389	7	69	606,914,235
Total	6526	3596	13,788,312,108	3158	1734	11,858,743,564	5731	25,508	13,739,987,325	1058	4922	13,881,369,116

**Table 4 plants-10-00853-t004:** Summary of quantitative trait loci (QTL) associated with the heading, flowering, and maturity dates in four recombinant inbred lines (RIL) populations evaluated under conventional and/or organic management systems. Both chromosomes and physical map positions are based on the International Wheat Genome Sequencing Consortium (IWGSC) RefSeq v2.0.

Trait	QTL Name	Population ^1^	Management ^2^	Chromosome	QTL Location ^3^	Confidence Interval (Mb)	R^2^ (%): Conventional	R^2^ (%): Organic
Heading	QHd.dms-1D	PAC	Org	1D	1D:325876831-336302522	10.4		2.8
Heading	QHd.dms-2B	PAC	Con & Org	2B	2B:55797422-62651542	6.9	11.5	5.6
Heading	QHd.dms-3B	PCS	Org	3B	3B:45204750-49026001	3.8		5.8
Heading	QHd.dms-4A	PCS	Con	4A	4A:17686877-17787027	0.1	8.8	
Heading	QHd.dms-5A.1	PAC	Con & Org	5A	5A:60279363-76684980	16.4	9.6	3.7
Heading	QHd.dms-5A.2	PAC	Org	5A	5A:620541477-621338663	0.8		5.9
Heading	QHd.dms-5B.1	PAC	Org	5B	5B:349752769-354808447	5.1		4.1
Heading	QHd.dms-5B.2	PAC	Con	5B	5B:396826652-400681156	3.9	5.6	
Heading	QHd.dms-5B.3	PAC	Con & Org	5B	5B:574535307-577015908	2.5	16.2	19.3
Heading	QHd.dms-5B.4	PCS	Con & Org	5B	5B:639044446-653916583	14.9	8.1	1.8
Heading	QHd.dms-7D	PAC	Con & Org	7D	7D:73333549-83998578	10.7	16.6	10.9
Flowering	QFlt.dms-2A.1	PAC	Con	2A	2A:37479415-37904660	0.4	2.0	
Flowering	QFlt.dms-2A.2	PAC	Org	2A	2A:700598519-700903870	0.3		2.3
Flowering	QFlt.dms-2B	PAC	Con & Org	2B	2B:55797422-62651542	6.9	5.7	2.9
Flowering	QFlt.dms-3B	CAB	Con	3B	3B:432437212-432750764	0.3	7.9	
Flowering	QFlt.dms-5A.1	CAB	Con	5A	5A:582841379-583000992	0.2	8.2	
Flowering	QFlt.dms-5A.2	ACG	Con & Org	5A	5A:587346439-587412126	0.1	19.2	20.8
Flowering	QFlt.dms-5B.1	PAC	Con	5B	5B:333880729-349752769	15.9	9.0	
Flowering	QFlt.dms-5B.2	PAC	Con & Org	5B	5B:574535307-577015908	2.5	8.4	13.6
Flowering	QFlt.dms-7A	CAB	Con	7A	7A:24248719-25391676	1.1	8.6	
Flowering	QFlt.dms-7D	PAC	Con & Org	7D	7D:73333549-83998578	10.7	12.6	12.4
Maturity	QMat.dms-1A.1	CAB	Con	1A	1A:35556032-39776886	4.2	3.7	
Maturity	QMat.dms-1A.2	CAB	Con	1A	1A:399444508-426644827	27.2	3.7	
Maturity	QMat.dms-1A.3	CAB	Con	1A	1A:443059667-462755618	19.7	3.7	
Maturity	QMat.dms-3A.1	PAC	Con	3A	3A:392012081-406623288	14.6	2.9	
Maturity	QMat.dms-3A.2	PAC	Org	3A	3A:513855127-553098513	39.2		3.6
Maturity	QMat.dms-3B.1	PCS	Org	3B	3B:45204750-49026001	3.8		2.1
Maturity	QMat.dms-3B.2	CAB	Con	3B	3B:821149227-835005886	13.9	3.3	
Maturity	QMat.dms-4A.1	PCS	Con	4A	4A:16086950-17686877	1.6	2.9	
Maturity	QMat.dms-4A.2	CAB	Con	4A	4A:65464862-68459336	3.0	1.7	
Maturity	QMat.dms-4A.3	PAC	Con	4A	4A:580400959-582726391	2.3	2.5	
Maturity	QMat.dms-4B.1	PAC	Con & Org	4B	4B:30510315-31959109	1.4	5.5	4.0
Maturity	QMat.dms-4B.2	ACG	Con	4B	4B:569184188-599613837	30.4	11.0	
Maturity	QMat.dms-4D	CAB	Con	4D	4D:21025268-32965037	11.9	3.9	
Maturity	QMat.dms-5A.1	CAB	Con	5A	5A:569871147-570121744	0.3	2.0	
Maturity	QMat.dms-5A.2	ACG	Con & Org	5A	5A:587346439-587412126	0.1	3.7	16.7
Maturity	QMat.dms-5A.3	PAC	Con & Org	5A	5A:619697105-620189324	0.5	2.5	1.8
Maturity	QMat.dms-5A.4	PAC	Con & Org	5A	5A:689113847-692552481	3.4	3.7	3.0
Maturity	QMat.dms-5B.1	PAC	Con	5B	5B:559880753-560779737	0.9	4.5	
Maturity	QMat.dms-5B.2	PAC	Org	5B	5B:574535307-577015908	2.5		3.7
Maturity	QMat.dms-7A.1	PCS	Con	7A	7A:96812406-110000000	13.2	7.1	
Maturity	QMat.dms-7A.2	PCS	Con & Org	7A	7A:133489405-134226943	0.7	7.1	2.0
Maturity	QMat.dms-7A.3	PCS	Con	7A	7A:680676790-717910027	37.2	5.3	
Maturity	QMat.dms-7D	PAC	Con & Org	7D	7D:73333549-83998578	10.7	11.4	14.0

^1^ CAB: ‘Cutler’ × ‘AC Barrie’; ACG: ‘Attila’ × ‘CDC Go’; PAC: ‘Peace’ × ‘Carberry’; PCS: ‘Peace’ × ‘CDC Stanley’. ^2^ Con: Conventional; Org: Organic management. ^3^ QTL location starts with chromosome and the QTL confidence interval in base pairs.

## Data Availability

All relevant files are included in this article and its Appendix A files.

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
