# Peer review of "Physical Mapping of QTL in Four Spring Wheat Populations under Conventional and Organic Management Systems. I. Earliness"

_plants, 2021, doi:10.3390/plants10050853_

Round 1

Reviewer 1 Report

In this study, the authors mapped SNPs of four wheat RIL populations onto the IWGSC RefSeq v2.0 physical map to reconstruct physical and linkage maps and reanalyzed QTL of phenotypes evaluated in previous studies. The description of materials is clear, the methods applied to physical/linkage map reconstruction and QTL mapping are reliable. I have some minor comments as follows:

Line 138: change “On average each RIL headed 49-52 days, flowered 52-57 days, and matured 88-98 days” to “Among the four RIL populations, populations means of days to heading, days to flowering and days to maturity ranged from 49 to 52 days, 52 to 57 days and 88 to 98 days, respectively”.

Line 142: did you construct four independent genetic maps using 3158, 6526, and 1058 SNPs, respectively, or you integrated 3158, 6526, and 1058 SNPs into one genetic map? Please clarify.

Line 160: the authors found that “all QTL were population-specific” for the phenotypes evaluated in this study, which means that none of the identified QTL can be transmitted across populations and therefore cannot be used for between-population prediction. It might be interesting to discuss why those QTL are population/environment-specific.

Line 218: (1) When comparing QTL results with previous studies, the order of mapping populations should be in the same order as that described in the introduction section. (2) Is it possible to put QTL comparison results between the old and present studies in a supplemental table?

Line 410: how the top hits were determined should be described with more details, e.g. alignment length, percentage of identity, etc.

Lines 431-432: (1) What do you mean about “by dividing the physical positions in base pair (bp) to 105”. (2) The QTL scanning was based on the integrated linkage/physical map or you simply ran QTL mapping in each population separately and integrated all mapped QTL in the final physical map? Please clarify.

Table 1: the order of mapping populations should be in the same order as described in the introduction section.

Figure 1 was not cited in the main text

Author Response

We appreciated the constructive comments and accommodated all points, which is summarized below.

Line 138: Rephrased as suggested

Line 142: We constructed linkage maps and did QTL mapping independently for each population, which has been clarified in the methods section.

Line 160: We have added a new section and discussed the effect of genetic backgrounds and managements in QTL detection in the discussion section.

Line 218: We have reorganized CAB and ACG populations' order in the discussion section (without track change to avoid confusion with newly added sections). We have also added a new supplementary table (Table S4) that summarizes all previous published QTL based on genetic maps.

Line 410: We have rephrased the method section to highlight this point.

Lines 431-432: The division from bp to 10e5 is revised. We did QTL scanning independently using a physical map of each population. The final QTL results were presented on the combined physical map of all four populations.

Table 1: Order of CAB and ACG reorganized as suggested.

Figure 1 citation in the manuscript corrected (it was typed as Figure S1 before)

Reviewer 2 Report

QTL analysis has been conducted on four wheat RIL populations for three earliness traits using wheat reference genome. Results showed fifteen QTL were detected under both management systems and some loci harbor important wheat genes, which lay solid ground for further cloning and characterize these genes controlling earliness trait. This research would draw large attention from scientific peers. Well done.

Author Response

We thank you very much for recommending our manuscript for publication. We have taken the point on the possibility of cloning a few of the QTL with larger effect and added a phrase to highlight this point in the abstract and the conclusion sections.

Reviewer 3 Report

It is a common breeding aim for breeders to develop early maturing cultivars. Mapping earliness related QTLs forms the basis for marker-assisted selection during molecular breeding. In this manuscript, the authors carried out physical mapping of earliness related QTL in four spring wheat populations under conventional and organic management systems. The authors developed 4 recombinant inbred line (RIL) populations. They investigated heading dates, flowering dates and maturity duration in these RIL populations under either conventional or organic management systems. They then mapped and characterized QTLs associated with these three earliness traits using the International Wheat Genome Sequencing Consortium (IWGSC) RefSeq v2.0 physical map. They identified a total of 44 QTLs related to heading, flowering and maturity time. Among the 44 QTLs, 15 of them are commonly detected in both conventional and organic management systems, and the remaining 29 QTLs are specific to either the conventional or organic management. Generally, they carried out a considerable field and lab works and their analysis provide some useful information for interested readers to carry out their own studies. Some comments are shown below.
The authors may need to explain why they used both conventional and organic management systems. Besides herbicides and rotation, any differences between conventional and organic management?
As 29 out of 44 QTLs are the conventional or organic management specific, the authors should provide more detailed description on their difference in management. 
The authors may need to discuss why different QTL mapping results were obtained between two different management systems. Any statistical analysis has been carried out to evaluate if there exist statistical difference on heading, flowering and maturity duration between conventional and organic management?
Many QTLs were specific to either the conventional or organic management. If some of these QTLs are suitable candidates that can be used for marker-assisted selection, any suggestion on how to use them?
Minor correction:
Line 446: Fifteen of the 59 QTL were common in the two management systems,
Line 32: Fifteen of the 44 QTL were common to both conventional and organic management systems
Line 24 (a) 5731 SilicoDArT and single nucleotide polymorphism (SNP), “(a)” should be changed into “(i)”.
Line 114  (i) map and characterize QTL associated with heading, “(i)” should be changed into “(1)”.

Author Response

We appreciated the constructive comments. We have included a paragraph in the introduction section to provide the rationale as why we were interested in both conventional (high N) and organic (low N) managements. Planting time was an additional difference between the two managements, which is now included in the methods section. Nitrogen level is the most likely source of genetic variation between the two managements, which has been well documented in the literature. Several studies have reported the management specificity of QTL expression across different levels of nutrients (mostly N), which is like ours. The new section included in the discussion covers this point. We have also included Figure 2, which shows the relationship between the two managements on a trait basis., which shows a regression coefficient ranging from 0.71 to 0.86. Regarding the suggestion on their use, the 15 QTL detected in both managements are the ones that should be the priority. Those detected in each management be considered depending on the situation. We have included a sentence in the discussion and conclusion section to highlight how the QTL may be used for MAS. 

Line 446 vs. line 32: In Supplementary Table S2, there were (36 under conventional and 23 under organic) but 15 of them were counted twice (common to both managements). The 44 QTL were after excluding double-counted ones. To avoid confusion, we have deleted 59 and replaced it with 36+23.

Lines 24 and 114: Corrected.